# Adherence to COVID-19 Preventive Measures among Dental Care Workers in Vietnam: An Online Cross-Sectional Survey

**DOI:** 10.3390/ijerph19010481

**Published:** 2022-01-02

**Authors:** Tai Tan Tran, Thang Van Vo, Tuyen Dinh Hoang, Minh Vu Hoang, Nhu Thi Quynh Tran, Robert Colebunders

**Affiliations:** 1Faculty of Odonto-Stomatology, University of Medicine and Pharmacy, Hue University, Hue 530000, Vietnam; trantantai@hueuni.edu.vn (T.T.T.); hoangvuminh@hueuni.edu.vn (M.V.H.); tranthinhuquynh@hueuni.edu.vn (N.T.Q.T.); 2Institute for Community Health Research, University of Medicine and Pharmacy, Hue University, Hue 530000, Vietnam; vovanthang147@hueuni.edu.vn (T.V.V.); tuyenhoang@hueuni.edu.vn (T.D.H.); 3Faculty of Public Health, University of Medicine and Pharmacy, Hue University, Hue 530000, Vietnam; 4Global Health Institute, University of Antwerp, 2000 Antwerp, Belgium

**Keywords:** dental care workers, COVID-19 prevention measures, adherence, quality of life

## Abstract

An online cross-sectional survey using a “snowball” sampling method was carried out to assess the adherence to COVID-19 preventive measures among dental care workers (DCWs) during the pandemic. Six questions concerning the COVID-19 preventive guidelines issued by the Vietnam Ministry of Health were used to evaluate DCWs’ adherence to preventive measures at dental care clinics. The quality of life of DCWs was assessed using the WHO-5 questionnaire and was defined as low if the total score was less than 13 points. Factors relating to adherence to COVID-19 prevention measures of DCWs were determined by multivariate linear regression analysis. In total, 514 DCWs completed the questionnaire. A total of 37% DCWs rated their quality of life as low. Regression analysis suggested that older age, a better quality of life, living in an urban area, and training on COVID-19 prevention were associated with better adherence to COVID-19 preventive measures, while being a dentist and lack of personal protective equipment was associated with less adherence to COVID-19 preventive measures. The pandemic had a significant negative impact on the physical and mental health of DCWs. Therefore, specific national guidelines for the prevention and control of the spread of COVID-19 in dental facilities should be issued.

## 1. Introduction

The World Health Organization (WHO) declared COVID-19 as a global pandemic on 11 March 2020 because of its alarming spread and severity [1]. Until 28 November 2021, the total number of confirmed COVID-19 cases in the world was at 260,493,573, of which 5,195,354 people died [2]. Since the first reported case of domestic infection, Vietnam has experienced four waves of the COVID-19 pandemic spreading. On 3 October 2021, there were over 939,463 cases and 22,283 deaths nationwide [3]. The appearance of the COVID-19 Delta variant caused a large-scale spread of the disease in Ho Chi Minh city and other provinces [3].

According to a WHO report in September 2021, only 6643 healthcare workers (HCWs) were reported to have died of COVID-19 [4]. However, WHO estimated that the number of deaths among HCWs is much larger than officially reported and possibly reaches over 180,000 people. Therefore, increasing the efforts to protect HCWs through COVID-19 vaccination should be a top priority, especially in countries with low vaccination rates and high vaccination hesitancy [4].

The COVID-19 pandemic had a severe impact on various professions in society, especially DCWs who were directly exposed to the COVID-19 virus at medical facilities. The high risk of infection among HCWs is caused by close contact with patients diagnosed with COVID-19 but also with undiagnosed cases of COVID-19. Lack of personal protective equipment (PPE) increases this risk [5,6].

In addition to physical health impacts, the COVID-19 pandemic also has a huge impact on the mental health of medical staff, causing depression, anxiety, and insomnia [7]. A study in Da Nang city, during the second wave of the pandemic, showed an increase in the prevalence of stress among frontline HCWs [8].

Vietnam has a large number of skilled dentists that contribute to international dental tourism because they offer quality dental care at an affordable cost. These services provide numerous standard treatment technologies such as orthodontic treatment, implants, porcelain crowns, bridges, dentures, fillings, root canal, tooth extraction, and full-mouth restoration, etc. The current number of DCWs in Vietnam is unknown, but in 2013, it was estimated that there were 608 dentists, and 203 dental technicians in Vietnam. DCWs are at high risk of infection during dental procedures because they are in direct contact with aerosols and droplets from patients [9,10,11]. Therefore, adherence to infection control procedures is essential [12]. According to the WHO COVID-19 guidelines for HCWs, the prevention of dental problems and self-care should be a top priority [13]. Patients should receive dental advice through teleconsultations, and social media need to promote good oral hygiene [13]. The American Dental Association has released interim guidelines to minimize the risk of COVID-19 infection before, during, and after dental treatment [14]. The Vietnam Ministry of Health (MoH) issued a general guideline for the prevention and control of COVID-19 infection in medical facilities. These guidelines include recommendations for wearing PPE and face masks, hand hygiene and hospital sanitation, medical waste treatment, and COVID-19 quarantine and isolation measures [15,16]. However, in Vietnam, there is currently no specific guideline for COVID-19 prevention and control for dental care services. According to WHO recommendations, periodical dental check-ups should be postponed, but dental emergency treatment still needs to be performed with adequate safety measures. It is important that adherence to safety measures is monitored in the dental care settings. This study aimed to evaluate the adherence to COVID-19 preventive measures by Vietnamese DCWs and to assess the impact of the COVID-19 pandemic on their quality of life in Vietnam.

## 2. Materials and Methods

### 2.1. Study Design

A cross-sectional online survey was conducted among DCWs across Vietnam from 21 August to 9 September 2021.

### 2.2. Study Procedures

#### 2.2.1. Sample Size and Sampling Method 

Sampling was conducted using a snowball approach: persons who completed the online questionnaire were encouraged to send a survey web link to their personal contacts in all dental care services countrywide. We tried to reach as many DCWs as possible during a twenty-day period. All eligible entries recorded within this period were included in the study. Five hundred twenty-five DCWs responded to the survey but nine did not agree to participate in the study, and in two the answers were incomplete. Therefore only 514 (97.9%) responses were included in the data analysis.

Factors considered to be possibly related to adherence to preventive measures included socio-demographic characteristics, knowledge, and skills on COVID-19 prevention, and the impact of COVID-19 on the lives of DCWs (Figure 1).

#### 2.2.2. Impact of COVID-19 on the Lives of DCWs

Fears and worry about the DCWs health as well as their relatives’ health were measured on a 5-point Likert scale (1 = not worried/afraid, to 5 = extremely worried/afraid) [17]. The impact of the COVID-19 pandemic on DCWs income was measured on a 5-point Likert scale from 1 = not reduced to 5 = extremely reduced. Change in working time was measured on a 5-point Likert scale from 1 = working time reduced a lot to 5 = working time increased a lot. Difficulty in accessing food and food resources in the past week was measured on a 5-point Likert scale from 1 = not difficult at all to 5 = extremely difficult. Being stigmatized or discriminated by people due to working in an environment with a high risk of infection with the COVID-19 virus was measured on a 5-point Likert scale from 1 = never to 5 = very regularly. Quality of life of DCWs was assessed using the World Health Organization (WHO-5) 5-question assessment tool over the past two weeks: “I have felt cheerful and in good spirits”, “I have felt calm and relaxed”, “I have felt active and vigorous”, “I woke up feeling fresh and rested”, “My daily life has been filled with things that interest me”. Each question was scored from 0 = never to 5 = full time. Quality of life was scored on a scale from 0 to 25 points, where 0 was the lowest quality of life level and 25 was the highest quality of life level. A score below 13 points was defined as low quality of life [18,19].

#### 2.2.3. Adherence to COVID-19 Preventive Measures

As preventive measures, we took into account the guidelines of the Vietnam MoH on prevention and control COVID-19 in medical facilities (decision No.5188/QD-BYT) [15]. Adherence to preventive measures was assessed using 6 questions, covering the following aspects: wearing PPE, correct use of face masks during patient care, regular hand hygiene, cleaning and disinfection of surfaces during patient care, safe disposal of waste, and other procedures to prevent transmission of patient’s saliva, blood, and other body fluids. The response to each question was scored from 0 points = never to 4 points = always, and a total score was calculated (maximum score 24 points). Additionally, we asked participants to self-evaluate how difficult it was to adhere to these preventive measures, and this was also reported using a 5-point Likert scale (1 = not difficult at all, to 5 = extremely difficult).

### 2.3. Data Analysis

IBM SPSS software version 20.0 (IBM Corp., Armonk, NY, USA) was used for statistical analysis. Continuous variables are reported as means with standard deviation (SD), while categorical variables appear as percentages. A multivariable linear regression model was used to identify factors associated with adherence to COVID-19 preventive measures.

## 3. Results

### 3.1. Characteristics of Participants

Of the 514 respondents, 222 (43.2%) were male, and 292 (56.8%) were female (Table 1). The mean age was 33 years ± 8 years. Most DCWs had undergraduate (94.1%) and postgraduate education (78%). About 80% of DCWs lived in an urban area, and 88.1% worked in public hospitals and private clinics.

### 3.2. Training and Source of Information about COVID-19 Infection

Three hundred fifty-three DCWs (68.7%) received training to improve their knowledge and skills to prevent COVID-19 infection (Table 2). The training content included updates on the epidemic situation, transmission routes and COVID-19 infection prevention measures, how to assess and identify risk factors for COVID-19 infection, clinical COVID-19 manifestations, how to determine the COVID-19 infection status, and isolation regulations related to COVID-19. Main information sources included television and radio (81.5%), website of the MoH (81.3%), and social media (86.8%) (Table 2).

### 3.3. COVID-19 Vaccination among Dental Care Workers

A total of 309 (75.0%) DCWs received at least one dose of COVID-19 vaccine, but 124 (24.1%) were not yet vaccinated. Of those not yet vaccinated, 65 (52.4%) were willing to get vaccinated with any type of COVID-19 vaccine, while 50 (40.3%) only accepted the vaccine they considered the best (Table 3).

### 3.4. Adherence to COVID-19 Preventive Measures

DCWs scored high for adherence to COVID-19 preventive measures (total score 20.94 ± 2.90) and reported moderate difficulty in adhering to these measures (total score 2.33 ± 1.17) (Table 4). Among the difficulties encountered, 51% of respondents reported a lack of PPE, uncooperative patients (23.5%), overcrowded medical facilities (19.1%), and lack of COVID-19 prevention and control guidelines (16%) (Table 4).

### 3.5. Impact of COVID-19 on the Quality of Life of Dental Care Workers

Dental care workers’ income dramatically decreased due to the COVID-19 epidemic (4.04 ± 1.15); they were moderately and very worried about their own health (2.58 ± 1.25) and the health of their relatives (3.01 ± 1.29) (Table 5). DCWs working time decreased significantly (68.9%) (Table 5). One hundred ninety (37%) DCWs self-rated their quality of life as low, in particular DCWs from Ho Chi Minh city (Table 5).

### 3.6. Factors Associated with Adherence to COVID-19 Preventive Measures by Dental Care Workers

A higher quality-of-life score, COVID-19 infection prevention training, living in an urban area, and older age were factors positively associated with adherence to COVID-19 preventive measures. A profession as a dentist and lack of PPE at health facilities were factors negatively associated with adherence to COVID-19 preventive measures (Table 6).

## 4. Discussion

In 2020, Vietnam was recognized worldwide to be extremely effective in fighting the COVID-19 pandemic. However, the COVID-19 virus is constantly evolving, and new mutations appear, threatening the control of the pandemic worldwide. Since April 2021, the Delta variant caused a large wave of infection in Ho Chi Minh City and many provinces in Vietnam [20].

### 4.1. Adherence to Preventive Measures

During dental care, DCWs are at high risk of COVID-19 infection. Indeed, through the creation of saliva aerosols, pathogens will be dispersed from the oropharynx and oral cavity and spread by air and by contaminating surfaces [21]. As such, it is necessary to train DCWs on strict adherence to infection control practices (use of hand sanitizer, face masks, and maintaining social distancing measures), reduction in dental droplets, and ways to manage air quality in dental treatment rooms by reducing the use of air conditioners and improving air exchange [22,23]. However, 161 (31.3%) of the 514 Vietnamese DCWs who participated in this survey did not receive training to improve their knowledge and skills concerning COVID-19 preventive measures. In an online survey in China among orthodontists, orthodontic residents, and nurses, the majority of respondents (80.2%) were confident that they had sufficient knowledge about COVID-19, but most of them had less than half of the answers about COVID-19 correct [24]. The Vietnamese DCWs regularly or always adhered to COVID-19 prevention measures, while only 38.5% of dentists in Balochistan (Pakistan) used N95 masks [25].

DCWs in Vietnam obtained information about the pandemic from television and radio (81.5%), the MoH website (81.3%), and social networks (86.8%). In 2021 in Pakistan, sources of COVID-19 information reported by DCWs included social networks (46.7%), television (26.7%), newspapers (10.8%), and other sources (15.8%) [26]. In Australia, most dental students (89.6%) received information about COVID-19 from official governmental sources [27]. In a study in India, 83% of dental students were considered to have adequate knowledge about COVID 19, and almost 80% reported adhering to appropriate practices regarding COVID-19 [28].

In our study, older age, a better quality of life, living in an urban area, and training on COVID-19 prevention, were associated with better adherence to COVID-19 preventive measures. However, dentists did adhere less optimally than other DCWs. In India, it was reported that most dental practitioners had a low level of knowledge (38.1%), and no relationship was found between professional experience and level of knowledge [29]. However, results from a multinational study among DCWs documented an association between level of knowledge and experience about COVID-19 and adherence to preventive measures [30,31]. A similar finding was reported in a study among DCWs in Saudi Arabia [32].

Since the outbreak of the pandemic in Vietnam in April 2021, dentists were faced with many difficulties in dealing with the pandemic. Therefore, it is necessary to provide sufficient PPE in the dental treatment room and update the prevention instructions for DCWs [24,25]. In Saudi Arabia, dentists above the age of 45 years and with longer working hours and years of experience significantly were more adherent to preventive measures [33]. In Vietnam, people living in urban areas were more adherent [34], perhaps because cities are the main places of outbreaks. In order to ensure social distancing, reduce the overload on tertiary hospitals, and decrease the risk of infection, tele-dentistry should be used [35,36].

Similar to what has been observed in other countries, the COVID-19 outbreak in Vietnam revealed the inadequate cooperation of pandemic-related health services and insufficient access to PPE [37]. Attention should be paid to protecting frontline health workers by providing PPE and continuous competence training [38]. Furthermore, dental care clinics must setup a patient classification system based on dental conditions, body temperature, and TOCC (travel, occupation, contact, and cluster history), and implement strict working procedures to prevent the spread of COVID-19 during dental treatment [39].

In total 390 (75.9%) Vietnamese DCWs had received at least one dose of the COVID-19 vaccine, but 24.1% were still unvaccinated. Of the unvaccinated, 52.4% were willing to be vaccinated, but 40.3% only wished to receive the vaccine they preferred. In an Italian study, 18% of dentists opposed vaccination [40]. In an online survey in February among 529 dentists in Lebanon, 86% reported that they were willing to receive or already received a COVID-19 vaccine [41]. Only about 21% of Egyptian HCWs accepted to be COVID-19 vaccinated [42]; in a survey in the United States, only 64.6% of medical professionals accepted the COVID-19 vaccine [43]. Immunization against COVID-19 protects against a deadly disease and also helps to create the herd immunity needed to control this pandemic [44].

Vaccination of DCWs should be a priority due to their high risks of infection. Compared with high-income countries and certain other southeast Asian countries, Vietnam has a relatively low rate of vaccination against COVID-19. On 11 September 2021, Vietnam had injected more than 28 million doses of COVID-19 vaccine, of which nearly 23.2 million were as a first dose (reaching 44.3% of the population aged 18 and above) and nearly 5.1 million were as a second dose (reaching 9.5% of the population aged 18 and above) [45]. Thus, the proportion of adults that still need to be COVID-19-vaccinated is still large considering the target of herd immunity. However, it is expected that in 2022, 70% of the population in Vietnam will be vaccinated with two doses of a COVID-19 vaccine [46].

In September 2021, data provided to WHO by 119 mainly high-income countries revealed that about 40% of all HCWs received a full dose of COVID-19 vaccination [47]. It is clear that more effort is needed to increase the COVID-19 vaccination rate among HCWs and to reduce vaccine hesitancy [48].

### 4.2. Impact of COVID-19 Pandemic on the Lives of DCWs

Due to the COVID-19 pandemic, the income of the Vietnamese DCWs decreased dramatically, and DCWs experienced moderate to high level of anxiety concerning their own lives and their relatives’ lives. In a cross-sectional online survey among 312 dentists from India, USA, UK, and Iran, dentists reported that they suffered from stress and anxiety due to the shutdown of their private dental clinics; 72.5% were worried about financial loss during the COVID-19 pandemic, and 60% were worried about lack of PPE [49]. Thirty-seven percent of the Vietnamese DCWs considered their quality of life to be at a low level of quality. This proportion was higher in Ho Chi Minh City, where the outbreak was most dramatic. A study on COVID-19’s impact on private practice and academic dentistry in North America highlighted the anxiety among DCWs due to the high risk of occupational infection [11]. A study in Pakistan found that 75% of dentists were afraid of being infected with COVID-19, and 80.9% wanted to close their dental care services during the pandemic [25]. Similar to other countries, DCWs in Vietnam reported increased stress because of the worsening of their financial situation due to the closure of dental care services, decreased number of clients, and the rising cost of services [37,50].

Worldwide, the COVID-19 pandemic increased the rates of anxiety and depression among HCWs [51]. While HCWs play an important role in limiting the impact of COVID-19, their lives have been greatly affected by the pandemic [52]. Therefore, efforts are needed to improve their quality of life by minimizing the stress and psychological sequelae associated with COVID-19 and by improving social support [53].

To prevent COVID-19 transmission among DCWs and HCW in general, a comprehensive approach of combining vaccination, training on prevention skills, and psycho-social support at health facilities will need to be implemented [48].

### 4.3. Study Limitations

This study has several limitations. Only 514 DCWs participated in the survey and we do not know the number and the characteristics of the Vietnamese DCWs who did not participate in the survey. Most likely, DCWs working in urban centers were overrepresented in the sample, as they were the easiest to reach. This created a selection bias. Recall bias is also a possibility when participants were asked to recall events from the past. Moreover, online data collection may lead to errors because participants may not have understood the questions. However, to avoid this we created a hotline where participants were able to contact us for more explanation.

## 5. Conclusions

Most Vietnamese DCWs do adhere to COVID-19 prevention measures. However, insufficient access to PPE remains a problem. Their quality of life is affected, especially in Ho Chi Minh city, the site in Vietnam most affected by the pandemic. The impact of the pandemic on the physical and mental health of the DCWs is significant. Therefore, policies are needed to support DCWs and to ensure their physical and mental well-being.

## Figures and Tables

**Figure 1 ijerph-19-00481-f001:**
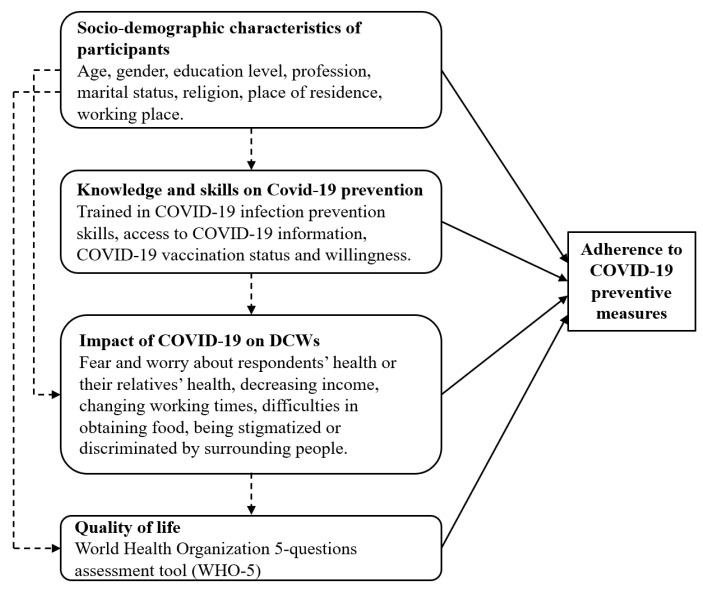
Conceptual framework of factors potentially related to adherence to COVID-19 preventive measures.

**Table 1 ijerph-19-00481-t001:** Characteristic of participants (*n* = 514).

Characteristic		*n*	%
Age (mean ± SD)		33 ± 8	
Min–Max		22–62	
Gender	Male	222	43.2
	Female	292	56.8
Education level	High school	30	5.9
	Undergraduate	287	55.8
	Postgraduate	197	38.3
Profession	Dentist	401	78.0
	Nurse	89	17.3
	Technician	16	3.1
	Other	8	1.6
Marital status	Single	208	40.5
	Married	294	57.2
	Divorced	7	1.4
	Separated	3	0.6
	Widow	2	0.4
Religion	No religion	415	80.7
	Buddhism	58	11.3
	Roman Catholicism	41	8.0
Place of residence	Urban	418	81.3
	Suburban/Rural	96	18.7
Working place	Public hospital	244	47.5
	Private hospital	43	8.4
	Private clinic	209	40.6
	Other	18	3.5

**Table 2 ijerph-19-00481-t002:** Training and source of information about COVID-19 prevention (*n* = 514).

Knowledge and Skills to Prevent COVID-19 Infection		*n*	%
Has been trained to improve their knowledge, skills to prevent COVID-19 infection	Yes	353	68.7
	No	161	31.3
Most accessible source of information about COVID-19	Television, radio	419	81.5
	Website of the MoH	418	81.3
	Electronic media	392	76.3
	Social media	446	86.8
	Relatives, friends, colleagues	394	76.7
	Other	8	1.6

**Table 3 ijerph-19-00481-t003:** COVID-19 vaccination among dental care workers.

**Received at least one dose of COVID-19 vaccine**	Vaccinated	390/514	75.9
	Not vaccinated yet	124/514	24.1
**Willingness to be vaccinated (*n* = 124)**	Yes, with any type of vaccine	65	52.4
	Yes, but only with the best vaccine	50	40.3
	Do not want vaccination due to health problems	6	4.8
	Do not want vaccination due to fear that the vaccine will affect their health.	1	0.8
	Other	2	1.6

**Table 4 ijerph-19-00481-t004:** Dental care workers adherence to COVID-19 preventive measures (*n* = 514).

Adherence to COVID-19 Preventive Measures		*n*	%
Measures Mean ± SD (Min–Max)	Wear PPE	3.21 ± 1.05 (0–4)	
	Use face mask correctly during patient care	3.64 ± 0.56 (0–4)	
	Practice hand hygiene correctly	3.59 ± 0.59 (0–4)	
	Clean and disinfect surfaces in patient care areas regularly	3.35 ± 0.75 (0–4)	
	Safe disposal of waste	3.61 ± 0.59 (0–4)	
	Apply procedures to prevent all transmission routes of COVID-19	3.54 ± 0.66 (0–4)	
	Total	20.94 ± 2.90 (4–24)	
Difficulty to adhere to epidemic prevention measures Mean ± SD (Min–Max)		2.33 ± 1.17 (1–5)	
Difficulties to adhere to preventive measures	Lack of PPE at health facilities	262	51.0
	Overcrowded health facilities	98	19.1
	Patients do not cooperate	121	23.5
	Lack of guidelines on COVID-19 prevention and control in medical facilities	82	16.0
	Others	12	2.3

**Table 5 ijerph-19-00481-t005:** Impact of COVID-19 on the quality of life of dental care workers (*n* = 514).

Impact COVID-19 Mean ± SD (Min–Max)	Fear and worry about respondents’ health	2.58 ± 1.25 (1–5)	
	Fear and worry about their relatives’ health	3.01 ± 1.29 (1–5)	
	Income decreased due to the impact of the COVID-19 pandemic	4.04 ± 1.15 (1–5)	
	Difficulties in obtaining food	2.41 ± 1.19 (1–5)	
	Being stigmatized or discriminated by others	2.20 ± 0.99 (1–5)	
Working time	Working time was greatly reduced	198	38.5%
	Working time reduced	156	30.4%
	No effect	73	14.2%
	Working time increased	56	10.9%
	Working time increased severely	31	6.0%
Total quality of life score Mean ± SD (Min-Max)	All participants	14.04 ± 5.77 (0–25)	
	Place of residence		
	Ho Chi Minh city	12.70 ± 5.66 (0–25)	
	Other places	14.46 ± 5.75 (0–25)	
Low quality of life	All participants	190/514	37.0%
	Place of residence		
	Ho Chi Minh city	57/122	46.7%
	Other places	133/392	33.9%

**Table 6 ijerph-19-00481-t006:** Factors associated with adherence to COVID-19 preventive measures by dental care workers * (*n* = 514).

Independent Variables	β (95% CI)	*p*
Gender: Male	0.42 (−0.11–0.94)	0.118
Age	0.03 (0.00–0.06)	0.048
Quality of life score	0.06 (0.02–0.10)	0.008
Profession as dentist	−0.85 (−1.47–−0.23)	0.008
Currently living in urban area	0.66 (0.03–1.29)	0.041
Trained in COVID-19 infection prevention skills	0.58 (0.04–1.12)	0.035
Lack of PPE at health facilities	−0.78 (−1.29–−0.27)	0.003

* Multivariable linear regression model. Dependent variable was the total score of adherence to COVID-19 prevention measures by DCWs.

## Data Availability

All answers were collected anonymously and stored in a Microsoft Form application. The datasets created and/or analyzed during the current study are available from the corresponding author upon reasonable request.

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
