# Peer review of "Adherence to COVID-19 Preventive Measures among Dental Care Workers in Vietnam: An Online Cross-Sectional Survey"

_ijerph, 2022, doi:10.3390/ijerph19010481_

Round 1

Reviewer 1 Report

Dear Authors, I’ve extensively read the present manuscript and I believe that, after some improvements, the manuscript is suitable for publication. From the methodological perspective I’ve no major concerns to report, just few suggestions, including some improvements in the and a revision of the English language text which in some parts is tough to read. The issue is very interesting, but there are a few suggestions to improve the manuscript:

  • Please follow the proper format of the journal`s rules. For example, use the same front and remove the unnecessary bold front.

  • In the Introduction section continuation of the sentence is not the proper way. Try to rearrange the sentence like- briefly describe the broad research area and then narrow down to your particular focus. 

  • Try to specify - why this research is important and describe more about the aim of the research.

  • In the results section (Table 2), No need to use n number in every line, please use n number in the headline only. For example Table 2 n= 514

  • Try to be very specific and precise through the article and revise English.

Author Response

Thanks for your very useful comments on the manuscript. 

Please see attachment file is our answers upon your comments and hope the revised manuscript is better   

Reviewer 2 Report

In order to have statistical significance, did the authors set a minimum number of respondents?
Is it possible to detail the results presented in table 6?
Are the work protocols, personal protection and patient protection the ones recommended by the WHO?

Author Response

Thanks for your very useful comments. We have addressed what you consulted in our answers of the attachment file

Reviewer 3 Report

1. The figure1, study design, needs to be modified. Please make corrections according to the analysis results.

2. The number of study subjects by dental care workers was not taken into account. In addition, the allocation of the number of study subjects for urban/sub-urban, rural is not appropriate. Please consider the limitations of this part in your analysis.

3. 'Quality of life' variables are usually analyzed as outcome variables. It is necessary to consider the use of 'quality of life' as an independent variable.

4. It is necessary to find a way to ensure that dental care workers comply with COVID-19 related prevention guidelines. Please describe this in the review section.

5. If there are reports of cases of COVID-19 infection in DCW during treatment, please describe it to the introduction.

6. It was said that the vaccination rate for dental care workers in Vietnam is higher than that of other countries, but a clear rationale for this content is needed.

Author Response

It's very grateful for your useful reviews on the manuscript. Please see the attachment file is what we addressed upon your comments
